# *Coffea canephora*: Heterotic Crosses Indicated by Molecular Approach

**DOI:** 10.3390/plants11223023

**Published:** 2022-11-09

**Authors:** Priscila Sousa, Henrique Vieira, Eileen Santos, Alexandre Viana, Marcela Boaechat, Fábio Partelli

**Affiliations:** 1Department of Plant Science, Universidade Estadual Norte Fluminense, Campos dos Goytacazes, Rio de Janeiro 28013-602, Brazil; 2Research Center, Universidade Estadual do Mato Grosso, Estudo e Desenvolvimento Agroambiental, Tangará da Serra 78300-000, Brazil; 3Department of Plant Science, Universidade Federal do Espírito Santo, São Mateus 29932-540, Brazil

**Keywords:** DNA markers, coffee, SSR markers

## Abstract

The genus Coffea comprises more than 100 species, of which the most commercially important are *Coffea arabica* and *Coffea canephora*. The latter is a self-incompatible plant with high natural genetic variability. The detection of polymorphism at the DNA level by molecular markers allowed significant progress with the selection of superior plants. The objective of this study was the molecular characterization of *C. canephora* using microsatellite markers. To this end, a population of forty-four *C. canephora* genotypes and one *C. arabica* genotype, was evaluated with 21 primers. These primers identified 61 alleles in the population and between 2 and 5 alleles per locus. The information index indicated a high level of polymorphism of the analyzed markers. According to the observed and expected heterozygosity, the genetic diversity in the population is high. The overall inbreeding coefficient of the population detected high heterozygosity and zero inbreeding within this population. Genetic diversity among the accessions was also evaluated by the unweighted pair-group method based on arithmetic averages (UPGMA). Six groups were formed based on Mojena’s cutting rule and three using the Bayesian approach. These results confirmed the existence of genetic diversity, genetic variability and a potential for selection in future breeding efforts involving the 45 genotypes studied.

## 1. Introduction

Of the more than 100 species of the genus Coffea, *C. arabica* L. and *C. canephora* Pierre ex A. Froehner account for approximately 99% of the global coffee bean production. Brazil is the world’s largest producer and second largest consumer of coffee. The production and demand of *C. canephora* has increased significantly due to its use in blends with *C. arabica*, to produce espresso and soluble coffee, owing to the high soluble solids content, high industrial yield and contribution to a full-bodied coffee [1].

In view of the gametophytic self-incompatibility, *C. canephora* is an allogamous species [2] with high natural genetic variability, cross-pollination and high genotypic diversity among plants in the same plantation [3,4]. In breeding programs, the availability of genetic variability is a basic prerequisite to ensure genetic gains with selection, and genetic diversity studies can serve as guidelines for the choice of parents in the development of clonal varieties [5].

In breeding programs in the past, plants were selected based on morphological markers, which are modified by environmental influence [6]. With the enhancement of molecular biology techniques, the selection of superior plants was significantly advanced by the use of molecular markers, for allowing the detection of polymorphism at the DNA level [7,8]. Molecular markers are more advantageous than phenotypic markers because they are free from environmental influence and also because they can be used at any stage of plant development [9].

These markers are already applied in studies on *C. canephora* breeding in the context of targeted high-throughput genotyping [10], genetic diversity characterization and species reclassification [11] or to evaluate the genetic diversity [12], population structure and genetic relationships [13].

Clonal cultivars of *C. canephora* must consist of a combination of genetically divergent genotypes to improve pollination efficiency and ensure successful crop production. In this sense, genetic diversity is essential to maintain productivity, quality and also raise the capacity of response to biotic and abiotic stresses [14]. Thus, the genetic characterization of *C. canephora* clones is fundamental in the planning of recombination strategies between genotypes and the composition of new cultivars, and SSR markers are an effective technique for these purposes. We raised the hypothesis that molecular markers such as SSR are able to detect differences at the DNA level allowing the distinction of genotypes without environmental influence.

Thus, the objectives of this research consisted of: (i) estimation of the genetic distance between plants or clones; (ii) genetic characterization, to measure and structure the genetic variability between genotypes; and (iii) identification of divergent plants for use as parents in coffee breeding programs.

## 2. Results and Discussion

### 2.1. Diversity Parameters via SSR

For this study, a total of 48 primers between SSR and EST were tested to identify the most divergent plants within this *C. canephora* population and thus analyze the molecular genetic diversity.

In the analysis of genetic diversity of the study population, 21 of the 48 tested primers produced amplified DNA fragments, which were selected for being polymorphic.

In evaluation studies by Baltazar et al. [12], on the genetic diversity of *C. arabica* accessions from several areas of the Philippines conserved in an ex situ collection, 19 SSR markers were used, 15 of which were polymorphic. The markers used in this study were also highly polymorphic, i.e., they can be used to detect differences between genotypes based on their genetic relationships.

The 21 primers detected 61 alleles in the population and 2 to 5 alleles per locus. Five alleles were detected at SSR-463 and SSR-1005, indicating greater diversity of these than of the other loci. Averaged across the entire data set, the study population had 2.9 alleles (Table 1). This low number of alleles may be associated with the clonal propagation of the population as well as the small population size.

However, the estimated allele frequency is strongly influenced by the number of plants in the population and tends to increase with increasing sample size. The reason is that in large samples, the chance of detecting rare, i.e., low-frequency alleles in the population is greater [15,16].

In studies by Kiwuka et al. [17], on the susceptibility to climate, reproductive potential and conservation of genetic diversity in native and cultivated *C. canephora* in Uganda, a mean of 4.7 alleles per locus in native populations was found. In comparison, a greater number of alleles was detected in the collections of cultivated genotypes. The relatively low allelic diversity found in the native population may be due to restricted gene flow between the closest populations, isolation by fragmentation, genetic drift, anthropogenic disturbance in those forests or mortality due to extreme stress conditions. These results were expected, since different populations and germplasm collections were analyzed in the above studies, while in this study only one clonal *C. canephora* population was evaluated, which explains the low number of alleles it contains.

In a study of digital DNA screening of microsatellite markers in Coffea sp germplasm, conserved in Costa Rica, Sanchéz et al. [18], found a mean of 4 alleles in Arabica coffee and 3.5 alleles in non-Arabica coffee (*C. canephora*, *C. excelsa* and *C. liberica*). These authors also evaluated 46 Arabica and 3 non-Arabica coffee genotypes and concluded that the mean number of alleles was influenced by the number of samples, their genetic base, number of microsatellites used and polymorphism level. This result is similar to ours, as the low allele frequency found in this study may be due to the low number of genotypes in the studied population.

In an analysis of the population structure and genetic relationships between 33 *C. arabica* genotypes and three diploid Coffea species (*C. canephora*, *C. eugenioides* and *C. racemosa*) from Brazil and Ethiopia, using 30 SSR markers, Da Silva et al. [13], identified a total of 206 alleles (mean of 6.9), i.e., high allele richness. Although in this study the number of alleles analyzed was lower, these are still a potential source of new alleles to be exploited in coffee breeding programs.

For the 21 microsatellite loci analyzed in this study, the information index (I) of 12 loci (57%) exceeded 0.5. The highest value was 1.038 for locus SSR-845, the lowest 0.184 for locus SSR-55^2^ and the mean across all loci was 0.58 (Table 1).

Studies by Botstein et al. [19], proposed that a locus can be classified as highly informative when the information index exceeds 0.5, moderately informative when between 0.5 and 0.25 and uninformative if lower than 0.25. According to this classification, the above mean value indicates a high level of polymorphism for the analyzed SSR-EST markers.

The genetic diversity of Arabica coffee in the Philippines was analyzed using SSR markers by Baltazar et al. [12], with a view to applications in breeding and selection, to guarantee the genetic stability of the coffee varieties. These authors found information indices between 0.140 and 0.609 and mentioned that most of the markers were moderate to highly informative, indicating their effectiveness in measuring the genetic diversity of the collection. Similarly, in this study, more than 50% of the loci were considered highly informative, which allowed an evaluation of the polymorphism at each locus of the study population with the forty-four genotypes of *C. canephora* and one *C. arabica* genotype. The index can be affected by the number of alleles and allele frequency at the locus.

The observed heterozygosity (Ho) ranged from 0.093 at locus SSR-1005 to 1 at locus SSR-744 (mean of 0.359). The observed heterozygosity is a measure of the frequency of heterozygotes in a studied sample and indicates the existence of genetic variation in the population, since each heterozygote has different alleles for a given gene, and the frequency of heterozygotes tends to be higher with higher genetic diversity.

According to Hardy–Weinberg expectations, the expected heterozygosity (He) ranged from 0.073 at SSR-55^2^ to 0.588 at locus SSR-08^1^ (mean of 0.34).

The observed heterozygosity was higher than expected at 17 loci (80%). When the Ho is greater than expected, this may suggest an excess of heterozygotes in the population in relation to the Hardy–Weinberg equilibrium model, De Moura et al. [20]. In the case of excess heterozygotes, there is no inbreeding, and the genetic diversity of the population is high, as the genotypes were previously selected in breeding programs.

In a study on the genetic structure and diversity of *C. canephora* in Upper Guinea, Labouisse et al. [21], found greater diversity in the Congolese than the Guinean group, with higher He and Ho values. They reported a mean He of 0.67 and mean Ho of 0.51 for the Congolese but a mean He of 0.48 and mean Ho of 0.34 for the Guinean group. The higher the number of alleles in a population, the greater the probability of it being heterozygous. The number of alleles found in this study, although rather low, may still contribute to the maintenance of the high diversity found in this population.

Antioxidant activity and stable free radicals in Robusta green coffee genotypes were analyzed by [22] and the authors stated that the natural reproduction of Robusta species generates highly heterozygous plants and populations with high genetic variability. Consequently, the characterization and exploitation of the genetic variability of this species might identify valuable genetic resources, for production systems or for breeding programs.

Analyzing the values of the fixation index (F) or inbreeding coefficient of the study population, the highest F values (0.947 and 0.522) were found at loci SSR-353 and SSR-1005. However, most values were lower and negative at 14 loci (66%) and the overall population mean was −0.035 (Table 1).

The fixation index is one of the most important parameters in population genetics, as it measures the balance between homozygotes and heterozygotes in populations, which can vary from −1 to +1. Values close to zero indicate random mating, negative values indicate excess heterozygosity and zero inbreeding for that locus in the study population, while high positive values indicate high inbreeding [23], causing the frequency of homozygotes to be higher than expected under H.W. equilibrium.

According to our results, the fixation index at one locus was very close to 1 (0.947). This may be related to the absence of alleles at that locus, caused by failure in the amplification of alleles, the occurrence of null alleles, or may also indicate the presence of inbreeding. According to Cruz et al. [24], the main effect of inbreeding is a decline in population heterozygosity. Another factor that may have raised the fixation index is the lower number of alleles at this locus, as according to Morgante et al. [25], the loci with the lowest number of alleles are probably located in genome regions where the transcription rate is lower than in the other regions.

Also, with regard to the fixation index, most values found were lower and negative, indicating heterozygosity excess at these loci. The general population mean was −0.035, i.e., zero inbreeding for this population. Another factor is that at the loci with negative values, heterozygosity was also greater than expected, which indicates that the alleles at these loci were not being fixed by inbreeding.

These data showed high rates of cross-pollination in the species under study as well as wide genetic diversity among the analyzed clones. Due to being a self-incompatible species, *C. canephora* receives pollen from several other coffee plants in its surroundings, maintaining the high heterozygosity and zero inbreeding, thus minimizing the risks of genetic drift and allele loss in the population. This result is extremely interesting from the point of view of productivity, showing that the establishment of homogeneous progenies for productivity and quality-related traits may be expected. Concomitantly, diversity is expected among the progenies, so that the best can be selected and cloned to ensure productivity and quality of the final product.

### 2.2. Relative Kinship and Estimates of Genetic Dissimilarity in C. canephora

Obvious kinship (K) was not detected in most individuals in this population (Figure 1). An approximate 76% of the pair-wise kinship estimates were between 0 and 0.09. About 23% between 0.1 and 0.3 and less than 1.5% of the pair-wise kinship estimates were from around 0.4 to 0.5 representing some familial relationships (Figure 1).

The genetic diversity among the accessions was evaluated using the average group linkage method (UPGMA), with the stopping rule of [26], resulting in six groups.

In genetic diversity studies with microsatellite markers, the most commonly used clustering method is UPGMA, which maximizes the cophenetic correlation coefficient and improves data reliability.

In this analysis, the cophenetic correlation index was 0.83, indicating a strong correlation between the dissimilarity matrix and the cophenetic distance matrix computed from the dendrogram. The cophenetic correlation estimate varies from 0 to 1; the higher this coefficient, the greater the representativeness of the dendrogram in relation to the genetic distance matrix. The diversity was confirmed by the formation of groups containing similar genotypes in the cluster analysis using the hierarchical UPGMA method (Figure 2).

Group I contained the largest number of plants, with 16 (35.55%). Group II contained 13 plants (28.88%), group III, 8 (17.77%), group IV, 6 (13.33%) and groups V and VI, 1 plant each (2.22% each). This amount of genotypes in the same group indicates that these plants share the highest number of alleles for the evaluated loci.

The seed-derived genotype and *C. arabica* genotype formed the fifth and sixth groups, respectively, with such high variability that they did not fit into any of the other groups. Groups with only one genotype were the most divergent in relation to the other groups. It is worth mentioning that both were used as a type of control. Genotype *C. arabica* was included in this study as control, and it was expected to have divergent genetic characteristics isolated from the *C. canephora* genotypes. Moreover, the fact that *C. arabica* formed an isolated group can confirm the effectiveness of the SSR–EST primers that are also used to distinguish different species of the genus Coffea.

The detection of divergent genotypes in genetically distinct heterotic groups of *C. canephora* populations is essential for the development of new cultivars of this crop. By selecting the most dissimilar genotypes for future crosses in breeding programs, the problems of genetic self-incompatibility in the species can be avoided.

The most genetically dissimilar genotypes, 122 and Arabica (0.71), had 13 alleles in common, at 16 of the 21 analyzed loci, although in this study Arabica was not used in crosses, but as a control.

Genotype Emcapa 02 was most dissimilar in relation to some genotypes, e.g., Z40, 700 and P1, with dissimilarity measures of 0.64, 0.63 and 0.61, respectively, and 16, 15 and 16 alleles in common.

The dissimilarity between the genotypes Peneirão and 122 and between Ouro Negro 2 and AT was 0.63 and 0.61, respectively; the first two had 16 and the latter 19 alleles in common.

These same genotypes were studied by [27], who reported a productivity of 97.10 bags ha^−1^ for Emcapa 02, 74.55 bags ha^−1^ for genotype Z40 and 89.19 bags ha^−1^ for genotype 700. The productivity of genotypes Peneirão and 122 was 81.07 and 99.22 bags ha^−1^, and genotypes Ouro Negro 2 and AT produced 74.70 and 128.93 bags ha^−1^, respectively. Genotypes 700 (Group III) and AT (Group IV) were not only dissimilar and highly productive, but also have an exceptionally low water requirement for mature coffee (less than 315 L) to yield one 60 kg bag of green coffee beans (Table 2).

Our data are promising since breeding targets the identification of genotypes with high productivity but also sufficient variability for selection, which was confirmed in the above-mentioned genotypes, with the greatest dissimilarities and high productivity.

The productivity of genotypes Bamburral and Z21 (Group I), A1 (Group II), L80 (Group III) and AT (Group IV) exceeded 100 bags ha^−1^ (118.08; 102.49; 108.19 and 128.93 bags ha^−1^, respectively) (Table 2). These high-yielding genotypes were allocated to different groups and can therefore be used in future crosses, targeting even higher productivity and variability.

The genotypes Emcapa 143 and Tardio V were the least dissimilar, in other words, the closest (0.06 dissimilarity, 39 common alleles). Most genotypes had shorter distances, probably due to the way in which these clones had been selected by farmers. In general, these producers all selected for the same agronomic characteristics of coffee trees, addressing high yield, plant height and disease resistance, among others. In this way, variability may have declined, and the clones may be somewhat similar. However, possible crosses will be proposed between plants from different groups, based on the greatest dissimilarity and high productivity.

The genotypes within a group are genetically similar, evidencing a certain pattern among most genotypes of the same group. This similarity was observed in groups I, II, III and IV, because in spite of the high genotypic diversity in this population, as mentioned above, some genotypes had a certain degree of genetic similarity and were therefore clustered in the same group.

The similarity between these genotypes can be explained by genetic drift that occurred when conilon was first introduced in the country, according to 30.

According to Fazuoli et al. [28], despite the high diversity, a large amount of seed from few plants was imported to establish the first plantations in Brazil, in the southeast region. The observed diversity is therefore still a minor sample of the natural diversity of the species; in addition, this variability has been decimated and may be at risk due to intense selection practiced by farmers over the years.

According to Cruz et al. [29] and Carmona et al. [30], the study of genetic divergence between accessions of a particular crop is essential to know and understand the genetic variability it contains. The resulting data are useful for the preservation and use of accessions, since genetically distinct and promising genotypes could be identified, making the grouping of these lines possible using statistical procedures, with a view to establishing homogeneity within and heterogeneity between the groups.

Genetic divergence analyses can be used to refine the selection of parents to be crossed. Crosses between highly divergent parents can maximize progeny heterozygosity and heterosis, thus increasing the chances of selecting superior elite plants. Clones with greater genetic dissimilarity can be selected for population formation, because by recommending a set of hybrids with greater genetic distances, the problems of genetic self-incompatibility of the species are minimized. In addition to the genetic distance, crosses between genotypes with a lower degree of kinship should be prioritized. As verified in Figure 1, the minority of peer-relatedness estimates were between 0.1 and 0.5.

### 2.3. Analysis of Genetic Structure

The Bayesian approach, based on the K criterion described by Evanno et al. (2005), used the alleles for inferences about the genetic structure of *C. canephora* genotypes and K = 3 was indicated as the most likely number of clusters (Figure 3 and Appendix A). In a Bayesian analysis of the genetic structure and diversity of *C. canephora* in Upper Guinea, Labouisse et al. [21] found K = 5.

An adherence probability of 70% that each genotype belongs to a certain group was assumed. The 45 genotypes evaluated were grouped as follows: 25 genotypes in group I (red), four in group II (green) and 16 in group III (blue) (Figure 4). Some plants have mixed probabilities, i.e., they have alleles in common, which belong to the three groups (Figure 3 and Appendix A).

Genotypes 37, 15, 32, 39, 2, 10, 29, 42, 7, 9, 27, 22, 20, 26, 3, 40, 5 and 28 have 100% adherence to the red group and genotypes 30, 4, 11, 36, 24, 19, 1, 43, 35 and 23 have 100% adherence to the blue group. This organization means that the genotypes in the red/blue/green group have sets of alleles that differentiate them for the set of markers used.

The plants 33, 16, 25, 34 and 8 have characteristics of all three groups, that means, less than 70% adherence to any of the three groups. Therefore, some plants are more likely to adhere to the red or the blue group, which could indicate that they share alleles. The SSR markers proved efficient in differentiating the different genotypes from the different locations, according to their genetic variability. The presence of genotypes with mixed probabilities can be explained by the genetic structure of the population, which comprises a small number of genotypes that share alleles with each other, due to their form of reproduction.

Using a Bayesian analysis, the groups were distinguished with regard to the analyzed loci and the plants that shared the same analyzed genomic regions were detected. This kind of clustering is promising as an orientation for crosses between *C. canephora* genotypes that belong to different groups, as they are genetically distant. It is important to combine these data with agronomic performance data of these genotypes, to select the plants with the greatest agronomic potential.

## 3. Materials and Methods

### 3.1. Experimental Area

The experiment was carried out in a plantation of 43 *C. canephora* genotypes, most of which had been selected by coffee growers in the region. Planting occurred in April 2014 in Nova Venécia, a county in northern Espírito Santo, Brazil, on a private property (lat. 18°66′23″ S, long. 40°43′07″ W; at least 199 m asl), where the mean annual temperature is 23 °C. The regional climate is tropical, characterized by hot and humid summers and dry winters, classified as Aw by Köppen (Alvares et al., 2013). The soil of the site was classified as a Latossolo Vermelho-Amarelo, distrófico, with clayey texture and wavy relief [27].

The genotypes were arranged in a randomized complete block design, with three replications, and each treatment consisted of seven plants of each genotype. The 44 genotypes (Table 3) were propagated from cuttings, planted at a spacing of 3 m between rows and 1 m between plants (3 × 1), equivalent to 3333 plants per hectare, and four stems per plant were left to grow. Two genotypes, identified as Clone 1 and Clone 2, were taken from another area to compare their genetic divergence with the above genotypes. For this reason, no yield-related data from previous scientific research was available. One *Coffea arabica* genotype of a cultivar (Catucaí 2SL) was also used, resulting in a total of 45 genotypes.

Management practices were applied according to the crop-specific technical guidelines and consisted basically of weed control with herbicides and brush cutters, preventive phytosanitary management, liming, fertilization and drip irrigation [27].

### 3.2. Sampling of Plant Material

Five light green (young) leaves were collected from the apex of each genotype (of one plant per genotype). The samples were immediately wrapped in aluminum foil, labelled and first frozen in liquid nitrogen. The samples were then transported on dry ice to the laboratory and stored at 80 °C until DNA extraction.

### 3.3. DNA Extraction

Genomic DNA was extracted and analyzed at the DNA Marker Sector of the Plant Breeding Laboratory of the Center of Agricultural Science and Technologies, State University of Northern Rio de Janeiro Darcy Ribeiro (LMGV/CCTA/UENF), Campos dos Goytacazes, RJ.

The frozen samples were ground with a pestle in crucibles with liquid nitrogen and stored in 2.0 mL microtubes. Genomic DNA was extracted by the method proposed by [34], with adaptations.

After extraction, the DNA was analyzed on 1% agarose gel with 1X TAE buffer (Tris, Sodium acetate, EDTA, pH 8.0), using a 100 bp Lambda (λ) marker (100 ng/μL^−1^) (Invitrogen, Waltham, MA, USA) and stained with a mixture of Gel RedTM and Blue Juice (1:1). The samples were photo-documented with the DNR MiniBIS Pro^®^ bio-imaging system. Based on the images, the DNA concentration was estimated in comparison with the 100 bp marker and the DNA samples were diluted to a working concentration of 10 ng/μL^−1^.

### 3.4. Primer Screening

The microsatellite and genomic primers for this study were selected from coffee cultivars for being polymorphic [35].

For the polymerase chain reaction (PCR), a total of 58 primer pairs were tested and used to amplify SSR and EST loci of *C. canephora* and *C. arabica*, within a temperature gradient of 50 °C to 60 °C. After screening, a set of 21 primer pairs was selected for the amplification reactions (Table 4).

### 3.5. Polymerase Chain Reaction (PCR)

A 35-cycle PCR amplification was carried out in an (Applied Biosystems™ Veriti™, Waltham, MA, USA) 96-Well Thermal Cycler, as follows: 94 °C for 4 min (initial denaturation); 94 °C for 2 min (cyclic denaturation); primer-specific temperature, in °C, for 1 min (annealing); 72 °C for 2 min (cyclic extension); 72 °C for 10 min (final extension); and holding step at 4 °C for unlimited time. The final volume of each sample was 13 μL, containing 2 μL DNA (5 ng/μL), 1.50 μL 10X Buffer (NH_4_SO_4_), 1.5 μL MgCl_2_ (25 mM), 1.5 μL dNTPs (2 mM), 1 μL primer (R + F) (5 μM) and 0.12 μL Taq-DNA polymerase (5 U/μL) (Invitrogen, Carlsbad, CA, USA).

The PCR products were diluted at a ratio of 6 μL sample to 18 μL buffer E of kit DNF 900, and subjected to capillary electrophoresis (Fragment Analyzer, AATI, Santa Clara, CA, USA), where amplified fragments between 35 and 500 bp long were separated at a resolution of approximately 2 bp. Each run lasted 2 h 20 min at 8 kW.

### 3.6. Statistical Analysis of Molecular Variables

The data of the 21 amplified SSR and EST markers were converted into numerical codes for each allele per locus. This numerical matrix was constructed by assigning values from one to the maximum number of alleles per locus, as described below: one locus with three alleles in homozygous form is represented as 11, 22 and 33 (A1A1, A2A2 and A3A3) and 12, 13 and 23 for those in heterozygous form (A1A2, A1A3 and A2A3). Based on the numerical matrix, three indices were tested: unweighted index, weighted index and the Smouse and Peakall index [23]. Based on the highest cophenetic correlation, the weighted index was used, and the analyses were run on GENES software [36].
(1)$$\overset{L}{\underset{j-1}{\sum }}p_{j}c_{j}$$
where:

*p_j_* = a_j_/A: weight associated with locus *j*, determined by:

a_j_: total number of alleles at locus *j*

A: total number of alleles studied
(2)$$\overset{L}{\underset{j=1}{\sum }}p_{j}=1$$

*c_j_*: number of common alleles between accession pairs i and i’.

The index represents similarity measures, while for cluster analysis, dissimilarity measures are recommended, defined by:D = 1 − S(3)

Once the distance matrix was calculated, dendrogram clustering was performed using the UPGMA method (Unweighted Pair-Group Method Average), on Mega software version 6 [37]. The optimal number of markers was estimated with GENES software [36] and the graph plotted with SigmaPlot software v14.

The genetic variability among the 45 genotypes was estimated with Genalex 6.5 software [23], based on the following parameters: number of alleles per polymorphic locus (NA), observed heterozygosity (Ho), expected heterozygosity (He), information index (I) and fixation index (ƒ). Relative kinship (K) among samples was calculated using 21 SSR markers by SPAGeDi software [38], and the Loiselle coefficient [39] was used to create the pair-wise kinship matrix (45 × 45).

### 3.7. Genetic Structure of the Population

To assess the structure of the 45 genotypes, a method based on Bayesian clustering algorithms was used, on STRUCTURE version 2.3.4 software [40]. To this end, we used the admixture model and independent allele frequencies, with a burn-in period of 250,000, followed by a final extension (Markov Chain Monte Carlo) of 750,000 iterations. Twenty simulations were run for each possible value of k (1–10).

The Δk statistical test was performed using Structure Harvester software and the Evanno criterion [41]. This criterion is based on the mean and standard deviation from the mean posterior probability (LnP(D)) values estimated at each of the 10 interactions per k. The Δki values were estimated as:Δki = ABS (ki + 1 − (2 × ki) + ki − 1)/standard deviation of Ki(4)
where:

I is the number of simulated groups, from i = 1 to i = 10; and ABS is the control module.

This Δk value was estimated for each k and the highest Δk selected. After choosing the optimal Δk, the simulation with the lowest LnP(D) value was chosen from a run of 10 simulations. In the resulting graph, each color represents a possible group of structured plants.

## 4. Conclusions

The SSR markers used in this study discriminated the *C. canephora* genotypes efficiently and were useful to make headway in breeding of conilon coffee. Genetic variability was confirmed in the evaluated population, which was structured into six distinct groups by UPGMA clustering and into three groups using a Bayesian analysis. Crosses between Bamburral and Z21 (Group I), A1 (Group II), L80 (Group III) and AT (Group IV) individuals are recommended in order to maintain the genetic variability of the population and high productivity. There is no kinship between the indicated genotypes.

## Figures and Tables

**Figure 1 plants-11-03023-f001:**
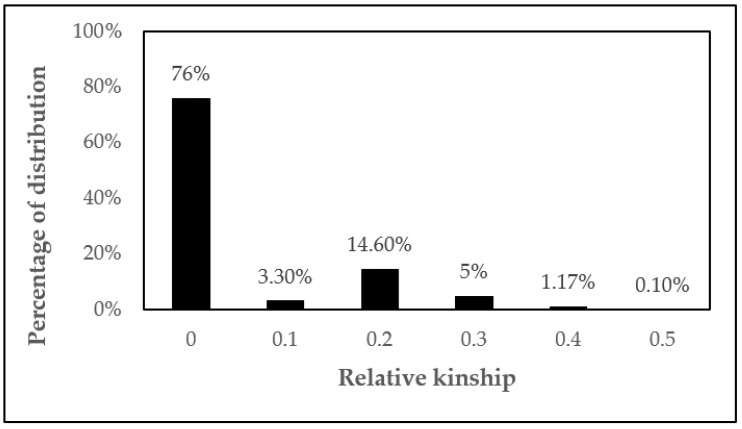
The distributions of pair-wise kinship coefficients for 45 genotypes of *C. canephora*.

**Figure 2 plants-11-03023-f002:**
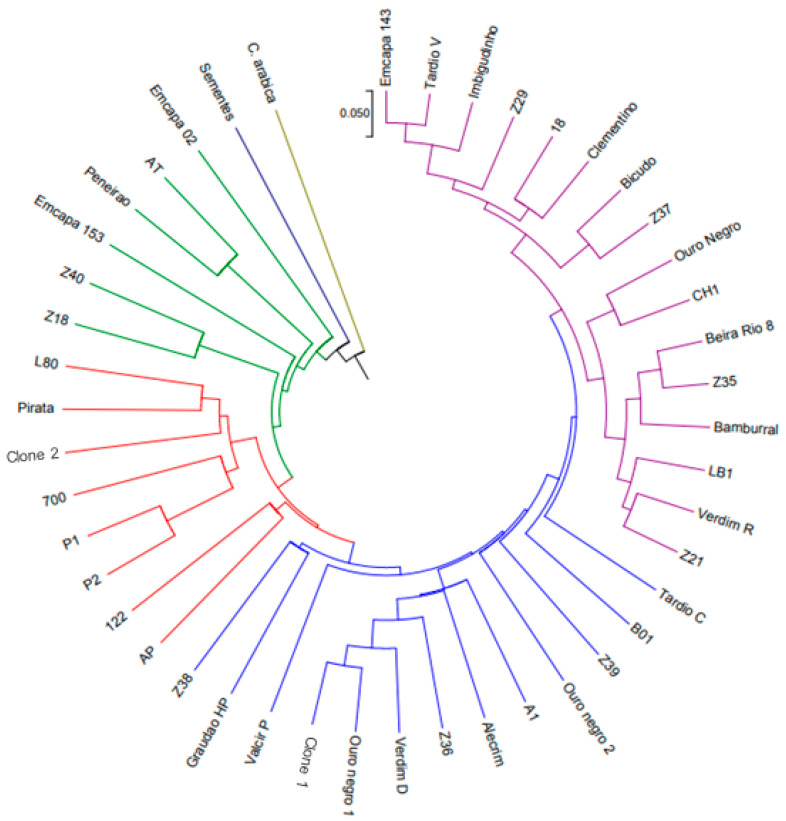
Dendrogram of genetic dissimilarity among 44 *Coffea canephora* genotypes and 1 *Coffea arabica* genotype, constructed using the UPGMA method, with SSR and EST markers.

**Figure 3 plants-11-03023-f003:**
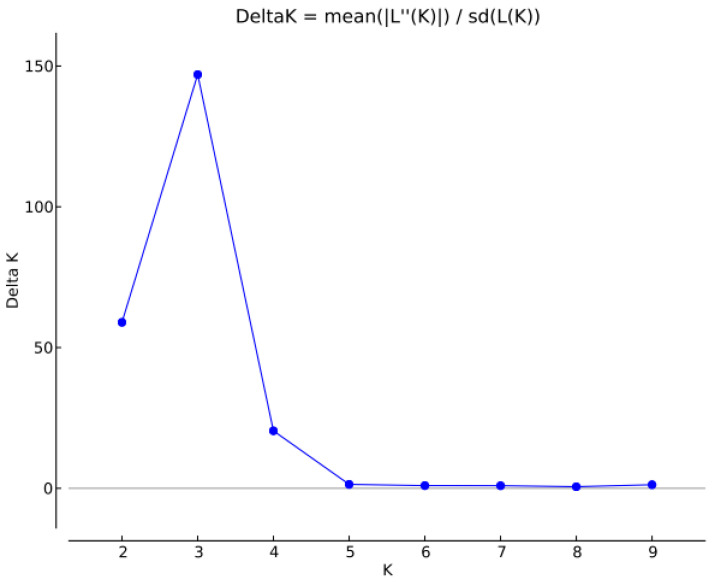
ΔK peak, indicating the optimal number of genetic clusters for Bayesian analysis with software Structure 2.3.4. UENF, Campos dos Goytacazes, RJ, 2021.

**Figure 4 plants-11-03023-f004:**
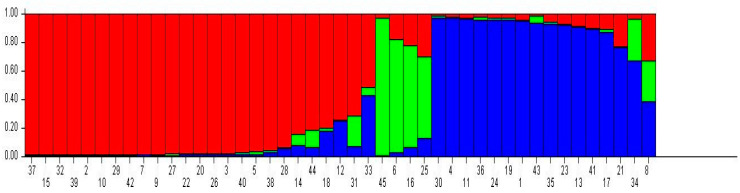
Bayesian inference clustering of 44 *Coffea canephora* and 1 *Coffea arabica* genotype. Genotypes are represented in the horizontal line and each genetic group is represented by a color. UENF, Campos dos Goytacazes, RJ, 2021.

**Table 1 plants-11-03023-t001:** Genetic parameters for the 20 SSR markers and 1 EST marker of the 44 analyzed genotypes. NA: number of alleles; I: information index; He: expected heterozygosity; Ho: observed heterozygosity; and F: fixation index.

Locus	NA	I	Ho	He	F
CaEST-010	2	0.413	0.289	0.247	−0.169
SSR-08^1^	3	0.987	0.5	0.588	0.15
SSR-30^3^	4	0.45	0.211	0.196	−0.076
SSR-34^3^	2	0.474	0.273	0.298	0.083
SSR-35^3^	2	0.65	0.024	0.457	0.947
SSR-37^3^	3	0.625	0.394	0.363	−0.085
SSR-43^3^	2	0.625	0.636	0.434	−0.467
SSR-46^3^	5	0.882	0.512	0.487	−0.05
SSR-48^2^	4	0.869	0.571	0.477	−0.198
SSR-49^3^	3	0.516	0.2	0.284	0.296
SSR-55^2^	3	0.184	0.075	0.073	−0.03
SSR-59^3^	2	0.264	0.148	0.137	−0.08
SSR-70^4^	2	0.528	0.395	0.344	−0.148
SSR-71^4^	2	0.253	0.14	0.13	−0.075
SSR-74^4^	3	1.003	1	0.609	−0.643
SSR-84^5^	4	1.038	0.432	0.553	0.219
SSR-87^5^	3	0.482	0.25	0.257	0.029
SSR-100^5^	5	0.462	0.093	0.195	0.522
SSR-106^1^	2	0.611	0.6	0.42	−0.429
SSR-114^6^	3	0.333	0.178	0.164	−0.086
SSR-119^3^	2	0.619	0.619	0.427	−0.448
Mean	2.904	0.584	0.359	0.34	−0.035

**Table 2 plants-11-03023-t002:** Mean yield in liters of coffee per 60 kg bag and productivity of the genotypes in the groups defined by microsatellite markers using the UPGMA clustering method.

Groups	Genotypes	Yield	Productivity
		Liter/Bag	Bag/ha
I	Emcapa 143	312.26	94.65
	Tardio V	329.95	69.84
	Imbigudinho	324.82	79.67
	Z29	366.61	70.15
	18	422.67	49.30
	Clementino	355.48	97.86
	Bicudo	366.21	93.09
	Z37	363.78	86.84
	Ouro Negro	346.39	75.24
	CH1	359.87	81.40
	Beira rio 8	439.72	61.81
	Z35	402.23	72.48
	Bamburral	357.23	86.60
	LB1	312.21	118.08
	Verdim R	398.32	82.35
	Z21	294.01	102.49
II	Tardio C	345.15	74.73
	B01	418.21	43.11
	Z39	339.92	95.48
	Ouro Negro 2	316.46	74.70
	A1	365.44	108.19
	Alecrim	341.46	54.37
	Z36	316.62	91.45
	Verdim D	336.65	90.43
	Ouro Negro 1	359.87	72.23
	Clone 1	-	-
	Valcir P	341.11	87.93
	Graudão HP	317.59	86.13
	Z38	372.53	74.94
III	AP	313.53	86.28
	122	330.03	81.07
	P2	316.7	92.32
	P1	332.5	93.47
	700	303.23	89.19
	Clone 2	-	-
	Pirata	379.63	76.22
	L80	364.68	105.53
IV	Z18	330.2	80.05
	Z40	356.18	74.55
	Emcapa 153	322.45	85.52
	Peneirão	326.6	99.22
	AT	303.49	128.93
	Emcapa 02	323.18	97.10
V	Sementes	348.06	70.32
VI	Arabica	-	-

Source: [27].

**Table 3 plants-11-03023-t003:** Identification of *Coffea canephora* and *Coffea arabica* genotypes used for molecular evaluation.

Sample Number	Name	Sample Number	Name
1	AP	24	Clone 1
2	Ouro negro 2	25	Z40
3	18	26	CH1
4	Ouro negro 1	27	Emcapa 143
5	Graudão HP	28	Verdim R
6	Peneirão	29	Tardio C
7	Ouro Negro	30	Clone 2
8	Z18	31	700
9	Clementino	32	Z29
10	Beira Rio 8	33	Emcapa 02
11	Verdim D	34	Pirata
12	Z39	35	A1
13	Bamburral	36	Z36
14	P2	37	Z37
15	Imbigudinho	38	B01
16	AT	39	Tardio V
17	Emcapa 153	40	Z21
18	Bicudo	41	Z35
19	Alecrim	42	LB1
20	Z38	43	L80
21	122	44	P1
22	Sementes	45	Arabica
23	Valcir P		

Note: Genotype 33 belongs to cultivar Emcapa 8111, genotypes 17 and 22 to cv. Emcapa 8131 [31], genotypes 28, 10, 13, 34, 35 and 09 to cv. ‘Tributun’ [32], genotypes 35 and 44 to cv. ‘Andina’ [33], and genotype 22 were derived from seed-propagated plants. The other genotypes do not belong to any cultivar.

**Table 4 plants-11-03023-t004:** Sequence of 21 microsatellite primer pairs used to analyze 44 *C. canephora* and 1 *C. arabica* genotype.

Locus	Sequence (5′3′)	at	Reference
CaEST-010	F: CTTCTTCATCCAACAACACG	54 °C	[35]
	R: TGCCATTCCACTGTGTCACT		
SSR-08^1^	F: CACTGGCATTAGAAAGCACC	55 °C	[35]
	R: GGCAAAGTCAATGATGACTC		
SSR-30^3^	F: ATGGGGCCAACTTGAATATG	55 °C	[35]
	R: CAGGGCATCTATCTACTTCTCTTT		
SSR-34^3^	F: GGAGACGCAGGTGGTAGAAG	55 °C	[35]
	R: TCGAGAAGTCTTGGGGTGTT		
SSR-35^3^	F: CTGGCATTAGAAAGCACCTTG	54 °C	[35]
	R: GCTTGGCTCACTGTAGGACTG		
SSR-37^3^	F: CAACACTATCTCTTGATTTTTCACT	53 °C	[35]
	R: CGTGCAAGTCACATACTTTACTAC		
SSR-43^3^	F: TTTTCTGGGTTTTCTGTGTTCTC	50 °C	[35]
	R: TAACTCTCCATTCCCGCATT		
SSR-46^3^	F: AATGAAGAAGAGGGTGGTG	53 °C	[35]
	R: CGAGGGTATTGTTTTCCAG		
SSR-48^2^	F: AGCAAGTGGAGCAGAAGAAG	56 °C	[35]
	R: CGGTGAATAAGTCGCAGTC		
SSR-49^3^	F: TGGAGAAGGCTGTTGAAACC	56 °C	[35]
	R: GGCGTGAAGCAAAAAGGTAT		
SSR-55^2^	F: GCAGGTATTTGAAGGATGAACC	56 °C	[35]
	R: GTGTAGGTGGTGCGATGTGT		
SSR-59^3^	F: CCAGCTCTCCTCACTCTTTTCA	58 °C	[35]
	R: GGTGGTGGAGGGGTAATAGG		
SSR-70^4^	F: GTAACCACCACCTCCTCTGC	59 °C	[35]
	R: TGGAGGTAACGGAAGCTCTG		
SSR-71^4^	F: GCTAAGTTCAATTGCCCCTGT	55 °C	[35]
	R: GGGTTAATTTGATTGCGTGA		
SSR-74^4^	F: TGGGGAAAAGAAGGATATAGACAAGAG	59 °C	[35]
	R: GAGGGGGGCTAAGGGAATAACATA		
SSR-84^5^	F: AAGTAGATTGGTGAAAGGGAAGC	57 °C	[35]
	R: TCCTTCATTTTCTCCTCTTGGTT		
SSR-87^5^	F: ATTCGACGACTCCAAAGCATA	58 °C	[35]
	R: CCTTGCTGGCCCTTCCTT		
SSR-100^5^	F: ACCCTTTACTACTTATTTACTCTC	50 °C	[35]
	R: ACATCCCCTTGCCATTTCTTC		
SSR-106^1^	F: CCCTCCCTCTTTCTCCTCTC	56 °C	[35]
	R: TCTGGGTTTTCTGTGTTCTCG		
SSR-114^6^	F: TAACAGAAGCACCAAAACC	53 °C	[35]
	R: TCTAAACCCACCTCACAAC		
SSR-119^3^	F: TTGCCATCATCGTTCATTCT	56 °C	[35]
	R: GCATAGTGTCGGTTGTGTTGTT		

## Data Availability

The study does not report any data.

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
