# Peer review of "Coffea canephora: Heterotic Crosses Indicated by Molecular Approach"

_plants, 2022, doi:10.3390/plants11223023_

Round 1

Reviewer 1 Report

Manuscript Plants ID: 195684

Title: Coffea canephora: heterotic crosses indicated by molecular approach

Autored by Priscila Sousa, Henrique Vieira, Eileen Santos, Alexandre Viana, Marcela Boaechat and Fábio Partelli

Dear Dr. Mrs. Biljana Skundric

Editor Plants

The study investigated the genetic diversity in 43 genotypes of Coffee species planted in Espírito Santo State, Brazil, which originated mainly from breeding programs, using 21 polymorphic microsatellite loci. The authors aimed to identify divergent genotypes to be used in future crosses in breeding programs for the species. The results showed that there is no inbreeding and there are some divergent genotypes in the populations, which can be used in future crosses. The conclusions are supported by the results. The article falls within the scope of the Plants Journal. However, I have some suggestions for improving the present manuscript.

However, I suggest that the authors estimate the relatedness (I suggest use the Loiselle et al. 1995) between the genotypes using, for example, the Spagedi program (Hardy and Vekemans 2002), as this is the most important information to be known in genetic improvement programs. Related genotypes may be divergent, but not suitable for use in crosses, as they may result in biparental inbreeding and inbreeding depression for survival and traits of economic interest.

Hardy O, Vekemans X (2002) SPAGeDI: a versatile computer program to analyze spatial genetic structure at the individual or population levels. Mol Ecol Notes 2:618–620. https://doi.org/10.1046/j.1471-8278

Loiselle BA, Sork VL, Nason J, Graham C (1995) Spatial genetic structure of a tropical understory shrub, Psychotria officinalis (Rubiaceae). Am J Bot 82:1420–1425. https://doi.org/10.2307/2445869

Minor corrections

Line 35. Where you read …high diversity…. What kind of diversity? Phenotypic?

Lines 36 and 44. Where you read … same plantation [3], [4]. In breeding programs…. I suggest .... same plantation [3,4]. In breeding programs.... Plese correct also line 44: [7,8] and 87 [15,16].

Lines 44-46. Where you read … Molecular are more advantageous than phenotypic markers because they increase gains with selection for the traits of interest, for being free of environmental influence and 45 also because they can be used at any stage of plant development [9].

It is unclear, because molecular markers are generally neutral and not associated with quantitative traits. Therefore, the selection for marker data does not generate genetic gains, unless they are linked to loci that determine the expression of quantitative characters. I suggest improving this sentence.

Line 51. Where you read … conilon coffee…. This is the first time the term is mentioned in the manuscript. Please specify what conilon coffee (robust coffee) means.

Lines 63-65. This sentence can be deleted.

Line 67. Where you read … C. canephora…. I suggest .... C. canephora....

The scientific names of the species must be in italics. Please, also correct along the manuscript.

Line 75. Where you read … ex situ ... the correct is .... ex situ

Line 74. Where you read …studies of [12], I suggest ....studies of Baltazar et al. [12],....

Line 88. Where you read …studies of [17],…. I suggest ....studies of Kiwuka et al. [17],....

Line 99. Where you read …, [18] found . I suggest .... Sánchez et al. [18] found ....

Line 108. Where you read …[13] identified, I suggest ...Da silva et al. [13] identified, ....

Line 115. Where you read …Studies of [19], …. I suggest .... Studies of Botstein et al. [19], ....

Line 120. Where you read … by [12], …. I suggest .... by Baltazar et al. [12], ....

Line 167. Where you read …cording to [24], …. I suggest .... cording to Cruz et al. [24], ....

Line 175. Where you read …cording to [25], …. I suggest .... cording to Morgante et al. [25],....

Please, check and correct along the manuscript, such as lines 142, 149, 193, 231, 258, 264, 284, 294, 302, 309, and 312.

Lines 134-135. Where you read … The He is determined based on the allele frequencies in the sample, assuming that the population is in equilibrium. This sentence can be deleted because it is very well known by those who work with population genetics.

Lines 139-140. Where you read … In case of excess heterozygotes, the genetic diversity of a population is high …. I suggest .... In case of excess heterozygotes, there is no inbreeding…

You could say more, such as mentioning that this happened because the genotypes were previously selected in breeding programs.

Lines 164-166. Where you read … According to our results, the fixation index at one locus was very close to 1 (0.947). This may be related to the absence of alleles at that locus, caused by failure in the amplification of one of the alleles, or may also indicate the presence of allele inbreeding. …. I suggest .... According to our results, the fixation index at one locus was very close to 1 (0.947). This may be related to the absence of alleles at that locus, caused by failure in the amplification of alleles, occurence of null alleles, or may also indicate the presence of inbreeding.

Lines 166-170. Where you read … According to [24], the main effect of inbreeding is a decline in population heterozygosity. Excess of homozygotes in the population indicates the occurrence of inbreeding and genetic drift, and one of the main inbreeding effects in a population is a decreasing frequency of heterozygous genotypes in the following generations.

Dears, there is no inbreeding for the average population. Inbreeding occurred at six loci, but mainly at two loci. This may be due to the presence of null alleles. You can check the occurrence of null alleles and correct the estimates for such alleles, using, for example, the INEST program (Chybicki IJ, Burczyk J (2009) Simultaneous estimation of null alleles and inbreeding coefficients. J Hered 100:106–113.)

Line 349. Where you read … in the region… Which region?

Line 362. Where you read … Coffea arabica..... I suggest .... Coffea arabica....

Lines 369-371. Where you read … (Giles et al., 2019; Partelli et 369 al., 2020), genotypes 35 and 44 to cv. 'Andina' (Martins, Partelli, Golynski, et al., 2019; Partelli et al., 370 2019),.... I suggest .... [36 = Giles et al., 2019, or 2018?; 27=Partelli et al.,  2020 or 2021? Please, check.), genotypes 35 and 44 to cv. 'Andina' [34, 27=Partelli et al., 2019 or 2021? Please, check.),....

Line 375. Where you read…. plant per genotype.. or plant per clone?

Lines 421 and 437. Where you read … (Peakall and Smouse, 2012)…. I suggest .... [23]....

Please, also correct the lines 434 (Kumar et al. 2009) [It is not in the references] and 435 [39].

Lines 446 and 451. Where you read … Ho: observed heterozygosity.. and He: expected heterozygosity... They can be excluded as Ho and He were defined above. In fact, it is not necessary to present how the Ho, He and F indices were estimated, as this is very well known to anyone who works with population genetics.

Line 478. Where you read … Coffea canephora…. I suggest .... C. canephora....

References

Check all references. The scientific names of the species must be in

Reviewer 2 Report

Please find the attached review report.
